# Ultrasound-guided internal branch of superior laryngeal nerve block on postoperative sore throat: A randomized controlled trial

Li Zhipeng[1]◉, He Meiyi[2]◉, Wang Meirong[2]‡, Jiang Qunmeng[2]‡, Jia Zhenhua[2], He Yuezhen[2], Zhang Jinfang[1]*, Liu Chuiliang[2]*

1 Key Laboratory of Orthopaedics & Traumatology, The First Affiliated Hospital of Guangzhou University of Chinese Medicine The First Clinical Medical College, Guangzhou University of Chinese Medicine, Guangzhou, China, 2 Department of Anesthesia, ChanCheng Center Hospital, Foshan, China

◉ These authors contributed equally to this work.
‡ These authors also contributed equally to this work.
* zhangjf06@gzucm.edu.cn (ZJ); 476140516@qq.com (LC)

**Data Availability Statement:** All relevant data are within the paper and its Supporting Information files.

## Abstract

### Introduction

Ultrasound-guided internal branch of the upper laryngeal nerve block (USG-guided iSLN block) have been used to decrease the perioperative stress response of intubation. It is more likely to be successful than blindly administered superior laryngeal nerve blocks with fewer complications. Here, we evaluated the efficacy of USG-guided iSLN block to treat postoperative sore throat (postoperative sore throat, POST) after extubation.

### Methods

100 patients, aged from 18 to 60 years old, ASA I~II who underwent general anesthesia and suffered from the moderate to severe postoperative sore throat after extubation were randomized into two groups(50 cases per group). Patients in group S received USG-guided iSLN block bilaterally (60mg of 2% lidocaine, 1.5ml each side), whereas those in group I received inhalation with 100 mg of 2% lidocaine and 1mg of budesonide suspension diluted with normal saline (oxygen flow 8 L /min, inhalation for 15 minutes). The primary outcome were VAS scores in both groups before treatment (T0), 10 min (T1), 30 min(T2), 1h(T3), 2 h (T4), 4h(T5), 8h(T6), 24h(T7), and 48h(T8) after treatment. The secondary outcome were satisfaction scores after treatment, MAP, HR, and SPO2 fromT0 to T8. The adverse reactions such as postoperative chocking or aspiration, cough, hoarseness, dyspnea were also observed in both groups.

### Results

Patients in group S had significantly lower VAS score than that in group I at points of $T_1 \sim T_6$ (P < 0.01). HR of group S was lower than that of group I at points of $T_1 \sim T_2$ and T4 (P < 0.05), and MAP was lower than that of group I at points of $T_1 \sim T_3$ (P < 0.05). Satisfaction scores of group S were higher than that of group I (P <0.05), In group S, 2 case (4%) needed

**Funding:** The authors received no specific funding for this work.

**Competing interests:** The authors have declared that no competing interests exist.

to intravenous Flurbiprofen Injection 50 mg to relieve pain; in group I, 13 cases (26%) received Flurbiprofen Injection. 2 case of group S appeared throat numbness after treatment for 3 hours; 2 patients have difficult in expectoration after treatment recovered after 3hour. No serious adverse events were observed in both groups.

## Conclusion

Compared with inhalation, USG-guided iSLN block may effectively relieve the postoperative sore throat after extubation under general anesthesia and provided an ideal treatment for POST in clinical work.

## Introduction

Postoperative sore throat (POST) is one of the common complications after extubation in patients under general anesthesia with tracheal intubation [1]. It is associated with mucosal damage and edema of the pharynx and larynx, for example, tracheal mucosa edema caused by excessive pressure of the cuff. Most POST have self-healing ability, but it could increase the discomfort of patients, prolong hospitalization time of patients [2, 3], and even affect the mortality rate [4, 5]. Many studies have been designed to find out possible approaches to reduce the occurrence of POST. Although, the occurrence of POST can be reduced by choosing a smaller type of endotracheal tube, softening tube, and using appropriate pressure of tracheal cuff, it is still an inevitable problem.

The medial branch of the superior laryngeal nerve accompanied by the superior laryngeal artery passed through the thyrohyoid membrane and divided into many small branches to the pharynx, epiglottis, and the sensory nerve of the laryngeal mucosa above the glottic fissure. Blocking the internal branch of the superior laryngeal nerve (iSLN) can achieve the anesthetic effect of the root of tongue, epiglottis, and laryngeal mucosa above glottis fissure [6]. Several studies have also shown that the blockade of the internal branch of the superior laryngeal nerve can be used during fiberoptic bronchoscopy, laryngoscopic surgery, conscious intubation of difficult airway to reduce the hemodynamic change caused by airway stimulation and provide better practical condition for operation [7–9]. Although others have reported bilateral iSLN block may reduce the severity of POST in assisted laryngoscopic surgery, we are not aware of any high quality randomized, controlled trials that have investigated the effectiveness of ultrasound guided internal branch of superior laryngeal nerve block (USG-guided iSLN block) on pcostoperative sore throat as compared with atomization inhalation, as treatment for POST.

We conducted a randomized clinical trial involving patients with POST to evaluate the efficacy and safety of ultrasound-guided superior laryngeal nerve block on postoperative sore throat. Our primary objective was to assess the efficacy of this method to relieved the POST. Our secondary objectives included satisfaction score of patient, hemodynamic response and adverse events during treatment. We hypothesized that administration of USG guided block of the iSLN would provide greater pain relief in patients with POST as well as more blunt the hemodynamic response after treatment, than does treat with atomization inhalation alone.

## Method

### Study population

The study was approved by the Medical Ethics Committee of Foshan Chancheng Central Hospital, and informed consent was signed by patients and their families. The register number was

ChiCTR1800015007. The ethics committee approved the study on Feburary 7, 2018. We registered this study on Feburary 28. Patients were enrolled in the study from June 12, 2018, to June 6, 2019. Participants were recruited for the study through advisement from Chancheng central hospital and the research was took place in PACU. Participants were enrolled if they fulfilled the following criteria: men or women between 18 and 60 years; ASA I or II; POST after extubation with tracheal intubation less than 4 hours; vital signs were stable after the operation in all patients. Patients with any of the following conditions were excluded: long-term smoking history; preoperative long-term throat discomfort and chronic laryngitis; patients who underwent oropharyngeal and neck surgery; patients who experienced difficult intubation and multiple intubations; patients who were unable to understand or cooperate with pain score; patients with long-term opioids, antipyretic analgesics or hormone treatment; patients who were allergic to local anesthesia. Because the syndrome is always self-healing, we follow-up the patients in 3days. All patients gave written informed consent before participation in this study. There were no important changes to methods after trial commencement.

## Randomization and blinding

Participants were enrolled to the study by the administrator if they had POST and signed the consent, They were randomly assigned in a 1:1 ratio (50 cases in each group) to receive atomization inhalation (I group) or received iSLN block (S group) using a computer-generated table of random numbers by the administrator who don't take part in the treatment. The envelope concealed the random numbers was opened by the anesthesiologist who performed the punctuation. The iSLN block was performed under the guidance of ultrasound. The allocation of participants was performed by an independent researcher at each clinical site who was not involved in outcome assessment. Patients in both groups were treated in the postanesthesia care unit and blinded to which treatment they would receive. The outcome assessors, data collectors, and statisticians were blinded to group allocations during the study (**Fig 1**).

## Interventions

All patients presented to operation room on the day of surgery after overnight fasting of 8 hours. After entering the operation room, the patients were given 8 ml/kg.h lactated Ringer's solution intravenously. MAP, HR, and SPO$_2$ were monitored routinely. Sufentanil (0.35 ug/kg), Propofol (2 mg/kg), and cis-atracurium (0.2 mg/kg) were given intravenously for anesthesia induction, and then general anesthesia was performed by tracheal intubation. No.7 endotracheal tube was used for women and No. 7.5 tube for men, respectively. All patients were successfully intubated at one time. The intubation was performed by the anesthesiologists who had over three years of standardized training. Narcotrend was used to monitor the depth of intraoperative anesthesia, while inhalation of sevoflurane (2%-3%), intravenous injection of Sufentanil (0.015 μg·kg-1·h-1) and cis-atracurium (5 mg/h) were used to maintain a certain depth of anesthesia. Following postoperative successful resuscitation, patients were sent to the post-anesthesia care unit (PACU) after extubation.

Patients were provided with mask oxygen inhalation at a concentration of 40%, ECG, HR, MAP, SpO2 were monitored routinely. When the patient is fully awake, the visual analogue scale (VAS) was used to evaluate the degree of sore throat, and a digital scoring method (0–10) was used to describe the degree of pain (0 was painless, 10 was the most severe pain).

USG guided block of the iSLN: For performing the block, patients were positioned supine, with the neck extended. A high-frequency (6–13MHz) ultrasound probe (Sonosite, USA) was placed over the submandibular area with a longitudinal orientation (**Fig 2**). The greater horn of the hyoid bone and thyroid cartilage were identified, which are hyperechoic signals on

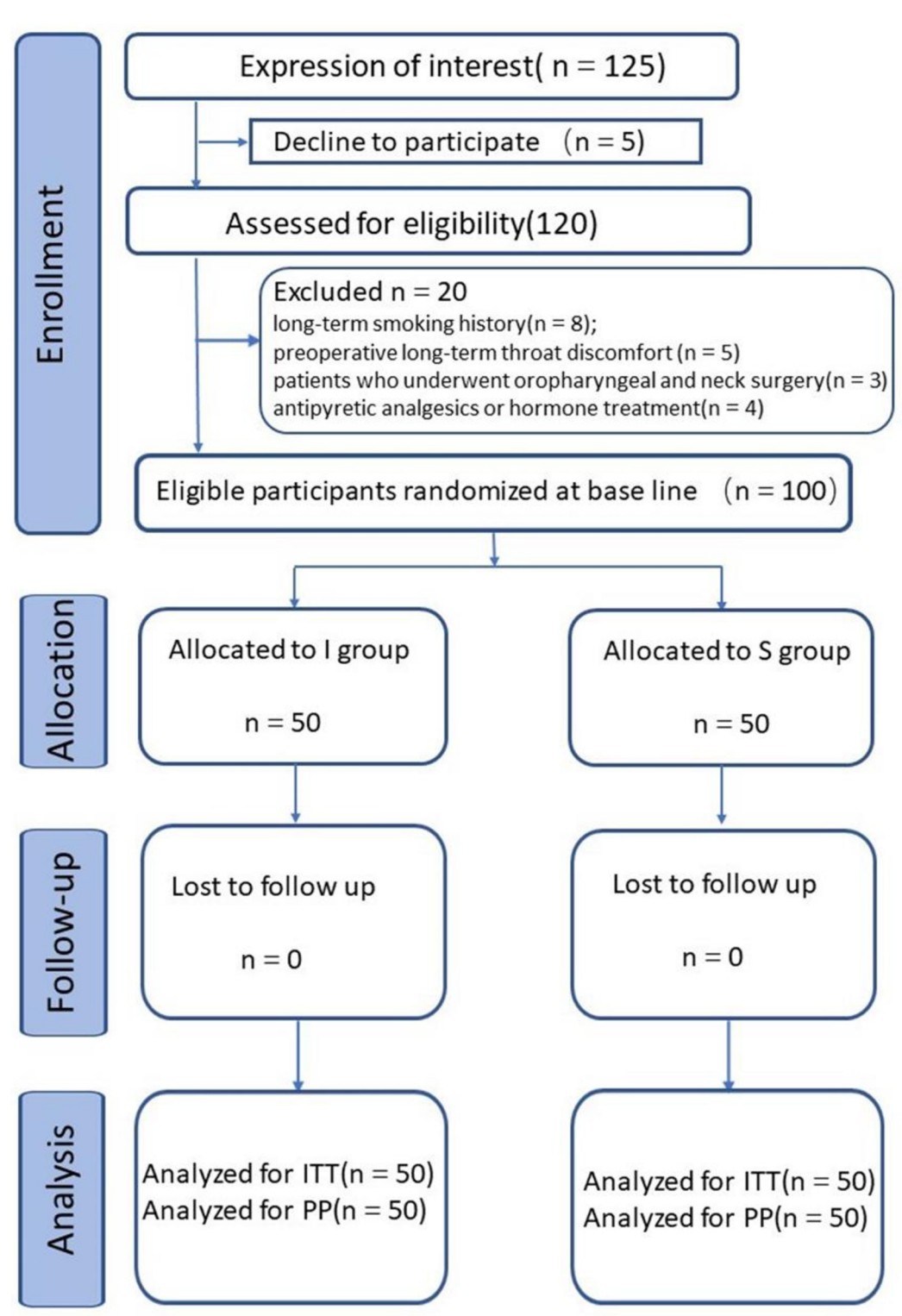

**Fig 1. CONSORT diagram of participant flow.** ITT = intention to treat; PP = per protocol.

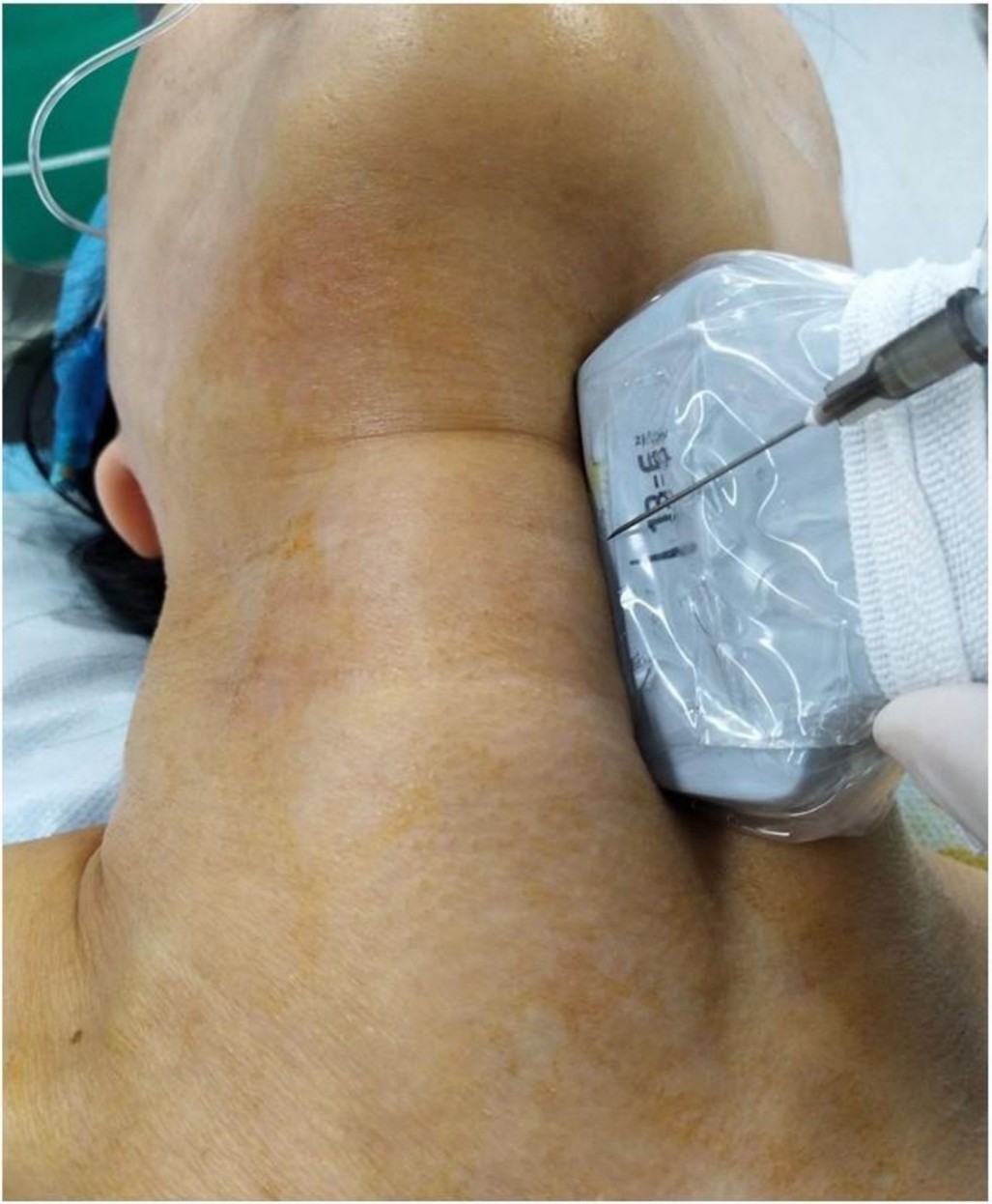

**Fig 2. The ultrasound transducer was placed between the hyoid bone and thyroid, the out plane technique was used.**

sonography. The thyrohyoid muscle and thyrohyoid membrane between these two structures and there was a hyperechoic mass between the detected two structures, which was the internal branch of superior laryngeal nerve (Fig 3). The block was performed using a 24gauge 1 inch needle with a 5 ml syringe that was filled with lidocaine 2%. An out-of-plane method was used to inject 2% lidocaine (1.5 ml each side) bilaterally, followed by local compression and observation for 5 minutes.

Patients in Group I were treatment with 2% lidocaine (100 mg) + Budesonide suspension (1 mg) diluted with normal saline to 10 ml (oxygen flow of 8 L/min). and mouthing-containing aerosol inhalation for 15 minutes. They inhale atomized gas with a nebulizer for

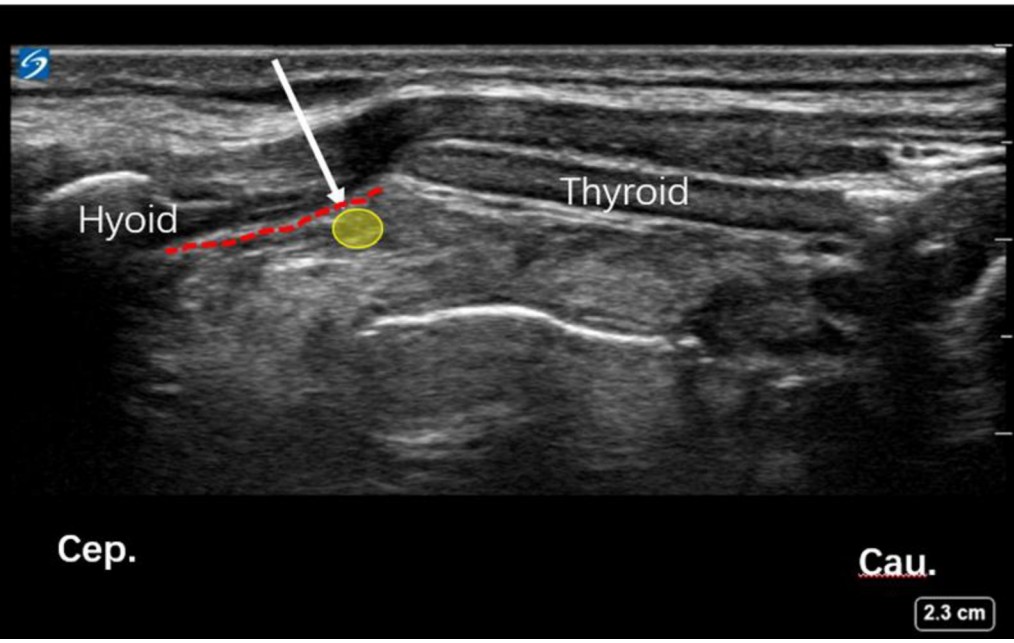

**Fig 3. Ultrasound image of iSLN block.** (A) Hyoid bone and thyroid cartilage were marked in the ultrasound image. (B) The red dotted line and yellow circle represented thyrohyoid membrane and superior laryngeal nerve respectively. (C) The white arrow represented the needle path.

15minutes. Patients didn't know other people's treatment method because they were separated into independent space by curtains.

Patients were sent to the ward of hospital after treatment, the anesthetists who was blinded to the treatment were assigned to follow-up and collect the results of treatment.

## Measures

**The primary outcome were VAS scores in both groups before treatment (T0), 10 min (T1), 30 min(T2), 1h(T3), 2 h(T4), 4h(T5), 8h(T6), 24h(T7), and 48h(T8) after treatment.** The VAS scores were recorded because that they are directly relative to POST. After 30 minutes of treatment, 50 mg of Flurbiprofen ester was given via intravenous injection to the patients with VAS$\geq$4 points to relieve pain, and the total dosage was recorded within 48 hours.

MAP, HR and, SPO 2 were recorded from T0 to T8, and adverse reactions were observed at the same time as the secondary outcome. Meanwhile, patients' satisfaction scores with treatment were also recorded 48 h after treatment. Two hours after the treatment of sore throat, patients were instructed to drink 20 ml of water to observe the occurrence of choking, aspiration and regurgitation, newly onset of hoarseness, and dyspnea.

## Statistical analysis

A detailed statistical analysis plan was made before the study was completed and any analyses were performed. An independent statistician who was unaware of the group assignments performed all the analyses. The software named IBM SPSS statistics 24.0 was used for the analysis. For VAS scores 2 hours after treatment according to previous study [10], a minimal clinically important difference of 10 is recommended and commonly used.

We calculated that a sample size of 49 patients in each group would give the study 80% power to detect a 20-point greater improvement in VAS scores in the S group than in the I

group at a two-sided significance level of 0.05. The calculation was operated on the website http://powerandsamplesize.com.

Age, BMI, intubation time, patient satisfaction score, VAS score of sore throat, blood pressure, and heart rate were continuous quantitative normally distributed data expressed as means and standard deviations (SD). Quantitative discrete data were expressed as median and range. Qualitative nominal data e.g. incidence of complications were expressed as percentage. Among them, age, BMI, intubation time, and patient satisfaction score were analyzed using Student's t-test, while inter-group comparison of VAS score of sore throat, blood pressure and heart rate was performed by repeated measures ANOVA. Greenhouse-Geisser's correction was applied when the Mauchly's test of sphericity was not followed, and multivariate analysis of variance was used to compare Group I and Group S at each time points. Chi-square or Fisher's exact tests were used as appropriate to compare qualitative data. A P-value<0.05 was considered statistically significant.

## Results

### General data

125 patients were interest to the research from January to August in 2019, 5 patients declined and 20 patients were excluded. The trial was stopped when enough patients were included to the study. 100 patients with moderate to severe sore throat after extubation under general anesthesia were enrolled in the study. There was no significant difference in gender, age, weight, height, BMI, and intubation time before treatment between the two groups (Table 1).

### VAS score at different time points between the two groups

VAS score of sore throat at T1~T6 in Group S was lower than that in Group I, with a significant difference between the two groups, but there was no significant difference between the two groups at T7~T8. The analgesic effect of sore throat in Group S was significantly better than that in Group L at T1-T6, whereas there was no significant difference between the two groups at the rest of time points (Table 2).

Besides, the results of repeated measures ANOVA indicated a significant trend in VAS through the study (F = 44.603, P< 0.05) and the two groups had significantly different trends (F = 68.566. P < 0.05).

### Hemodynamics at different time points between the two groups

The MAP of Group S was lower than that of Group I at T1-T3 (Table 3), HR was lower than that of Group L at T1, T2, and T4 (Table 4), and higher than that of Group I at T4. However,

**Table 1. Demographic details of patients.**

| Variable | | I group | S group | $X^2$ or t | aP |
|---|---|---|---|---|---|
| Female sex | (male/female) | 14/36 | 18/32 | 0.735 | 0.391 |
| Age (y) | (mean±SD) | 36.1±7.8 | 37.9±6.9 | -1.18 | 0.963 |
| Height | | 162.8±7.2 | 162.2±6.2 | 0.466 | 0.266 |
| Weight | | 59.5±8.0 | 59.2±7.3 | 0.532 | 0.17 |
| BMI | (kg/m²) | 22.4±2.8 | 22.5±2.7 | -0.126 | 0.676 |
| ASA grade | | 6/44 | 5/45 | 0 | 1 |
| Intubation time | (min) | 96.8±18.9 | 102.6±24.3 | 0.102 | 0.739 |

Abbreviations: BMI, body mass index(calculated as weight in kilograms divided by height in meters squared);ASA, American Society of Anesthesiologists

[a]P values were calculated using chi-square test or t test.

**Table 2. VAS scores of both group.**

|  | I group | S group | t | P |
|---|---|---|---|---|
| $T_0$ | 6.1±0.7 | 6.2±0.7 | -0.576 | 0.566 |
| $T_1$ | 4.4±1.0 | 2.0±0.8[a] | 13.186 | 0 |
| $T_2$ | 2.8±0.8 | 0.9±0.8[a] | 11.907 | 0 |
| $T_3$ | 2.2±0.8 | 0.8±0.7[a] | 9.27 | 0 |
| $T_4$ | 1.5±0.9 | 0.6±0.8[a] | 5.379 | 0 |
| $T_5$ | 1.0±0.7 | 0.4±0.5[a] | 4.729 | 0 |
| $T_6$ | 0.6±0.5 | 0.3±0.5[a] | 2.671 | 0.009 |
| $T_7$ | 0.3±0.4 | 0.2±0.4 | 0.907 | 0.367 |
| $T_8$ | 0.3±0.4 | 0.2±0.4 | 0.464 | 0.644 |

Abbreviations: Compared with group L, a represented P < 0.05; group I: inhalation group; group S: medial branch block of superior laryngeal nerve group; T0: immediately before treatment; T1: 10 minutes after treatment; T2: 30 minutes after treatment; T3: 1 hour after treatment; T4: 2 hours after treatment; T5: 4 hours after treatment; T6: 8 hours after treatment; T7: 24 hours after treatment; T8: 48 hours after treatment. P values were calculated using t test.

there was no significant difference between the two groups at the remaining time points. Repeated measures ANOVA indicated a significant trend in MAP and HR through the study (F = 49.516, P = 0.000; F = 50.427, P = 0.000) but the Group L didn't have significantly different trends compared with Group I through the study (F = 2.231, P = 0.138; F = 0.882, P = 0.350).

## Satisfaction and adverse reactions

The satisfaction score of Group S (3.4±0.7) was higher than that of Group I (3.0±0.7), and the difference was statistically significant (t = -3.070, P = 0.003). 2 cases (4%)intravenous injection of Flurbiprofen Axetil (50 mg) was needed in this group. While 13 patients (26%) still needed 50 mg of Flurbiprofen Axetil intravenously to relieve pain after treatment ($\chi^2$ = 9.490, P = 0.002). Furthermore, in Group S, there were 2 case with throat numbness and discomfort for 2 hours after block, and 2 case with expectoration weakness. There were no chocking, aspiration and regurgitation, newly onset of hoarseness, dyspnea in both groups.

**Table 3. Comparison of MAP at different time points after operation between the two groups [(x±s), mmHg, n = 50].**

|  | I group | S group | t | P |
|---|---|---|---|---|
| T0 | 88.6±4.7 | 90.0±8.4 | -0.992 | 0.325 |
| T1 | 84.1±3.8 | 79.3±8.9 [a] | 3.495 | 0.001 |
| T2 | 87.7±4.6 | 79.4±7.3[a] | 6.826 | 0 |
| T3 | 84.9±4.1 | 77.6±6.4[a] | 6.895 | 0 |
| T4 | 75.5±4.1 | 77.7±7.5 | -1.78 | 0.078 |
| T5 | 75.8±3.8 | 77.1±5.6 | -1.264 | 0.209 |
| T6 | 77.0±4.1 | 78.4±4.19 | -1.59 | 0.115 |
| T7 | 76.3±3.6 | 77.3±3.5 | -1.445 | 0.152 |
| T8 | 76.6±3.8 | 77.9±4.2 | -1.673 | 0.097 |

Note: Compared with group L, aP < 0.05; group L: aerosol inhalation group; group S: medial branch block of superior laryngeal nerve group; T0: immediately before treatment; T1: 10 minutes after treatment; T2: 30 minutes after treatment; T3: 1 hour after treatment; T4: 2 hours after treatment; T5: 4 hours after treatment; T6: 8 hours after treatment; T7: 24 hours after treatment.
P values were calculated using t test.

**Table 4. HR comparison of two groups at different time points after operation [(x±s), times/min, n = 50].**

|  | I group | S group | t | P |
|---|---|---|---|---|
| $T_0$ | 89.6±7.0 | 89.6±8.9 | 0.05 | 0.96 |
| $T_1$ | 85.6±6.0 | 78.3±6.5[a] | 5.823 | 0 |
| $T_2$ | 83.6±5.9 | 78.7±6.3[a] | 4.008 | 0 |
| $T_3$ | 79.5±5.8 | 77.6±5.7 | 1.655 | 0.101 |
| $T_4$ | 81.2±5.3 | 78.3±6.4[a] | 2.56 | 0.014 |
| $T_5$ | 77.3±5.6 | 79.0±5.8 | -1.531 | 0.129 |
| $T_6$ | 74.5±6.5 | 76.6±5.1 | -1.787 | 0.077 |
| $T_7$ | 74.2±5.9 | 76.3±6.3 | -1.734 | 0.086 |
| $T_8$ | 74.3±6.3 | 76.3±6.0 | -1.698 | 0.093 |

Note: Compared with group L, aP < 0.05; group L: aerosol inhalation group; group S: medial branch block of superior laryngeal nerve group; T0: immediately before treatment; T1: 10 minutes after treatment; T2: 30 minutes after treatment; T3: 1 hour after treatment; T4: 2 hours after treatment; T5: 4 hours after treatment; T6: 8 hours after treatment; T7: 24 hours after treatment.

P values were calculated using t test.

## Discussion

Sore throat after extubation is one of the most common complications during the recovery of anesthesia. Previous studies have shown that the incidence of sore throat after extubation was 24%-70% in general anesthesia patients [2]. It is mainly related to injury of tracheal mucosa, mucosal inflammation and ischemia caused by endotracheal intubation [4]. Besides, intubation without the neuromuscular block agent, intubation with double-lumen endotracheal tube, or excessive pressure of the cuff may also increase the risk of POST [11]. Some POST patients may recover by themselves, but some patients could be affected by POST for a long time, which seriously decreases the satisfaction of patients with anesthesia. We have taken some treatment measures, such as lidocaine (intravenous injection, external use of gel and atomization inhalation), steroids (intravenous injection and atomization inhalation), non-steroidal anti-inflammatory drugs, intravenous injection of N-methyl-D-aspartate (NMDA) receptor antagonists (magnesium, ketamine) [12]. However, there is still no satisfactory treatment for those patients with the severe sore throat.

In this study, we provide evidence that ultrasound guided superior laryngeal nerve block could provide greater pain relief in patient, more blunt the hemodynamic response and higher satisfaction scores after treatment than does treated with atomization inhalation alone. We founded that VAS score of sore throat at T1~T6 in Group S was significantly lower than that in Group I. As expected, ultrasound guided superior laryngeal nerve block could alleviated the POST syndrome in patient. We also founded that the MAP and HR rate of Group S was more stable than that of Group I; the satisfaction score of Group S (3.4±0.7) was higher than that of Group I. (3.0±0.7), which indicated that this method could improve hemodynamic parameters and the satisfaction of patients.

Previous studies indicated that USG guided block of the iSLN before intubation can reduce the incidence of sore throat after extubation under general anesthesia [13]. However, these approaches reported in prior studies were carried out before intubation for prevention. The difference of our study is that approaches were applied for pain treatment instead of prevention. In our study, the internal branch of the superior laryngeal nerve was blocked under the guidance of ultrasound, the VAS score decreased significantly, the analgesic effect was faster, than that of atomization inhalation with hormone combined with local anesthetics.

The reasons of iSLN block to alleviate sore throat were as follows. Firstly The internal branch of superior laryngeal nerve innervates most of the mucosal sensory in the laryngopharynx above the glottic fissure, and involves branches of the sympathetic trunk and superior cervical ganglion. iSLN block can not only block noxious stimulation but also produce sympathetic nerve block (In this study, it was also found that MAP and HR in the early stage of ultrasound-guided bilateral iSLN block were significantly lower than those in the atomization inhalation group, which may be related to the alleviation of pain and block of sympathetic nerve resulting in expansion of throat blood vessels and reduction of edema. Secondly, local anesthetics may provide an analgesic effect, improve local blood supply and produce anesthetic effect on local tissues. In this study, a short-acting local anesthetic of lidocaine was selected for the experiment and proved as an efficient medicine.

The ultrasound-guided technique was used to block the internal branch of superior laryngeal nerve in this study. A vertical puncture can be achieved between the region when touching the greater horn of hyoid bone and the upper horn of thyroid cartilage. After breaking through the thyroid hyoid ligament, local anesthetics can be injected carefully. Previous study has demonstrated that ultrasound-guided localization was superior than application of anatomical localization, with shorter duration of intubation, better tolerance and hemodynamic stability, and higher degree of comfort intubation using fiberoptic bronchoscope in patients with difficult airway in patients during the operation. Anatomical localization required deep palpation of the hyoid bone, which can make patients uncomfortable, and the failure rate of block would be higher for patients with shorter or thicker necks. Manikandan S. et al. [14] applied ultrasound-guided iSLN block for awake tracheal intubation in patients with cervical posterior fixation. The authors placed the ultrasound probe in front of the cervical vertebra and used intraplanar puncture technique. The upper laryngeal artery was identified as a marker, and local injection was made near the superior laryngeal artery after the blood was withdrawn. Awake intubation was then carried out without the discovery of any complications after iSLN by injecting local anesthetics near the superior laryngeal artery. Furthermore, prior evidence has shown that iSLN can be identified and located accurately under the guidance of ultrasound [15, 16]. Many reports revealed that iSLN was difficult to visualize under ultrasound [17, 18]. It was found that ibSLN could not always be displayed under ultrasound scanning, but thyrohyoid membrane and the superior laryngeal artery penetrating through the thyrohyoid membrane with internal branch of superior laryngeal nerve could be easily visualized [19]; besides, when local anesthetics were injected into the surface of thyrohyoid membrane and the "space" in the medial region of the superior laryngeal artery, satisfactory anesthetic effects were also achieved [16, 17]. In our study, the high-resolution ultrasound imaging system was used to confirm the visibility of ultrasound-guided iSLN image. When iSLN can not be clearly imaged, the spatial structure around the superior laryngeal nerve was selected as a substitute marker. The greater horn of hyoid bone and thyroid cartilage were identified as markers by the longitudinal location of ultrasound. Extra-planar puncture or transverse location of the thyrohyoid membrane were used to track the intra-planar puncture of the superior laryngeal artery. It was a reliable and repeatable method to use the peripheral structure of superior laryngeal nerve as a localization marker, which was consistent with the spatial anatomical structure of superior laryngeal nerve and indicated a high success rate of the blockade [16].

In this study, there was 1 case of throat numbness and discomfort for 2 hours after nerve block. There were no adverse reactions such as choking when drinking, aspiration and regurgitation, newly onset of hoarseness, dyspnea, nausea and vomiting in both groups. iSLN block is a peripheral nerve block method, and its blocking effect will lead to abnormal throat sensation, weakened protective reflex, and possible inhalation of gastric contents. The puncturing process damages adjacent blood vessels or tissues at times, causing hematoma or peripheral

tissue injury. Vasovagal reactions caused by excessive neck operation can lead to hypotension and bradycardia [19]. Blockade of the external branch of the superior laryngeal nerve may lead to low voice and blockade of unilateral recurrent laryngeal nerve may result in hoarseness. It has been reported that hoarseness occurs in part after ultrasound-guided bilateral ibSLN block, and aphasia and dyspnea after bilateral recurrent laryngeal nerve block [20]. The introduction of ultrasound-guided technology can greatly reduce the occurrence of such complications. To reduce the possibility of nerve injury or blocking other nerves, it is suggested that operators should be trained strictly to improve puncture technique and be skilled in basic knowledge of anatomy and ultrasonography to avoid the occurrence of adverse reactions in operation Meanwhile, short-acting local anesthetics could increase block safety, and visualization of local anesthetic diffusion guided by ultrasound could reduce the injection volume and avoid blocking other nerves [21, 22].

## Limitation

This study still has some limitations that are described as follows. Firstly, we used a single concentration and a single dose of local anesthetics in this study, with further study required concerning the optimal concentration and dose of local anesthetics. Secondly, some patients with throat and neck surgeries and multiple intubations were excluded from this study, which remains to be explored in the future to the therapeutic effect of these special patients.

## Conclusion

Ultrasound-guided bilateral ibSLN block with local anesthetic of lidocaine has a faster onset and better analgesic effect than traditional atomization inhalation of hormone combined with local anesthetics for the treatment of POST. Besides, its analgesic effect does not fade with the effect regression of local anesthetics, associated with higher patients' satisfaction and fewer complications, which provides an alternative safe and effective method for clinical treatment of POST.

## Supporting information

**S1 Checklist. CONSORT 2010 checklist of information to include when reporting a randomised trial**∗**.**
(DOC)

**S1 Dataset. Data sets of the research.** De-identified datasets were uploaded.
(XLSX)

**S2 Dataset. SPSS document of the datasets.** SPSS Document of de-identified datasets.
(SAV)

**S1 Protocol.**
(DOC)

**S2 Protocol.**
(DOCX)

## Author Contributions

**Methodology:** Li Zhipeng, He Meiyi, Wang Meirong, Jiang Qunmeng.

**Software:** Wang Meirong.

**Supervision:** Jia Zhenhua, Zhang Jinfang.

**Writing – original draft:** He Yuezhen.

**Writing – review & editing:** Li Zhipeng, Liu Chuiliang.

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
