## [Decision Letter · Decision Letter 0]

21 Aug 2020

PONE-D-20-15831

Ultrasound-guided internal branch of superior laryngeal nerve block on postoperative sore throat

PLOS ONE

Dear Dr. Zhipeng Li

Thank you for submitting your manuscript to PLOS ONE. After careful consideration, we feel that it has merit but does not fully meet PLOS ONE’s publication criteria as it currently stands. Therefore, we invite you to submit a revised version of the manuscript that addresses the points raised during the review process.

I would like you to pay careful attention to the reviewers' comments in your response.

We look forward to receiving your revised manuscript.

Kind regards,

Ehab Farag, MD FRCA FASA

Academic Editor

PLOS ONE

Journal Requirements:

2) Thank you for submitting your clinical trial to PLOS ONE and for providing the name of the registry and the registration number. The information in the registry entry suggests that your trial was registered after patient recruitment began. PLOS ONE strongly encourages authors to register all trials before recruiting the first participant in a study.

(a) your reasons for your delay in registering this study (after enrolment of participants started);

(b) confirmation that all related trials are registered by stating: “The authors confirm that all ongoing and related trials for this drug/intervention are registered”.

Please also ensure you report the date at which the ethics committee approved the study as well as the complete date range for patient recruitment and follow-up in the Methods section of your manuscript."

3) Please provide additional details regarding participant consent. In the ethics statement in the Methods and online submission information, please ensure that you have specified (1) whether consent was informed and (2) what type you obtained (for instance, written or verbal, and if verbal, how it was documented and witnessed). If your study included minors, state whether you obtained consent from parents or guardians.

4) In your Methods section, please provide additional information about the participant recruitment method and the demographic details of your participants. Please ensure you have provided sufficient details to replicate the analyses such as: a) a description of how participants were recruited, and b) descriptions of where participants were recruited and where the research took place.

5) Please provide a sample size and power calculation in the Methods, or discuss the reasons for not performing one before study initiation.

6) To comply with PLOS ONE submission guidelines, in your Methods section, please provide additional information regarding your statistical analyses, including the name and version of the specific statistical software used for the analysis. For more information on PLOS ONE's expectations for statistical reporting, please see https://journals.plos.org/plosone/s/submission-guidelines.#loc-statistical-reporting.

7) Please amend either the title on the online submission form (via Edit Submission) or the title in the manuscript so that they are identical.

8) Please include captions for your Supporting Information files at the end of your manuscript, and update any in-text citations to match accordingly. Please see our Supporting Information guidelines for more information: http://journals.plos.org/plosone/s/supporting-information.

9) We note that you have indicated that data from this study are available upon request. PLOS only allows data to be available upon request if there are legal or ethical restrictions on sharing data publicly. For information on unacceptable data access restrictions, please see http://journals.plos.org/plosone/s/data-availability#loc-unacceptable-data-access-restrictions.

Reviewers' comments:

Reviewer's Responses to Questions

**Comments to the Author**

1. Is the manuscript technically sound, and do the data support the conclusions?

Reviewer #1: Yes

Reviewer #2: Partly

Reviewer #3: Yes

2. Has the statistical analysis been performed appropriately and rigorously? 

Reviewer #1: I Don't Know

Reviewer #2: No

Reviewer #3: No

3. Have the authors made all data underlying the findings in their manuscript fully available?

Reviewer #1: Yes

Reviewer #2: Yes

Reviewer #3: Yes

4. Is the manuscript presented in an intelligible fashion and written in standard English?

Reviewer #1: Yes

Reviewer #2: Yes

Reviewer #3: No

5. Review Comments to the Author

Reviewer #1: Why the comparison was between US guided versus non US guided internal branch of superior laryngeal nerve block

Cost of the technique and experiences and instruments needed is required to discuss

Please compare between both groups as regards length and type of the surgery done as it may affect results

Was aspiration score was done

Clarify or how long period the internal branch of superior laryngeal nerve block work

Reviewer #2: The objective of this study is to conduct a two-arm RCT to evaluate the comparative effectiveness of the USG-guided iSLN block (group S), versus the inhalation (group I) to treat POST. While the study objectives sound interesting, some shotcomings were observed, in regards to abiding by the CONSORT guidelines for conducting and reporting results of high-quality RCTs. A model paper to follow abiding by the CONSORT guidelines is below:

https://www.sciencedirect.com/science/article/pii/S0889540619300010

The study, however, is registered within the Chinese CT Registry, and is accessible via a ChiCTR number.

Methods:

An orderly manner for Methods reporting is suggested, following CONSORT guidelines, without repeating information, such as Trial Design, Participant Eligibility Crtieria and settings, Interventions, Outcomes, sample size/power considerations, Interim analysis and stopping rules, Randomization (details on random number generation, allocation concealment, implementation), Blinding issues, etc. The authors are advised to create separate subsections for each of the possible topics (whichever necessary), and that way produce a very clear writeup. Please find some particular comments:

(a) Randomization: For instance, the steps of sequence generation (methods used to generate random allocation sequence), allocation concealment (methods to implement random allocation sequences) and blinding should be made very clear. Note, allocation concealment and blinding are not the same thing, and should be reported separately. The trial staff recruiting patients should not have the randomization list. Randomization should be prepared by the trial statistician, and he/she would not participate in the recruiting. Also, was a (block) randomization performed? If block, then what's the block size? Those details are necessary. How was the allocation sequence generated?

(b) Sample size/Power: A statement on sample size/power in the manuscript is presented, but it is not clear what (statistical) test was used, the effect size, etc. Those need to be clearly stated.

(c) Study Design & Statistical Analysis: The study has a relatively short duration; I wonder how meaningful it is to evaluate the effects via the repeated measures ANOVA? Also, the statistical analysis entirely relied on Gaussian assumptions of the responses; was it guaranteed? Also after conducting ANOVA, how was multiplicity testing adjusted to compare between various time-points?

In general, what is the perspective of this study once the 48 hours has passed?

Reviewer #3: Li and colleagues present in this paper results of a formally correct study to test the efficacy of ultrasound-guided internal branch of superior laryngeal nerve block on postoperative sore throat. Although, the following considerations should be considered to strenghten the paper:

1. The population sample is calculated on the basis of VAS score at 2h (T4): only T4 should be considered as primary outcome for sample size calculation, the other timepoints should be considered as secondary outcomes.

2. The VAS score was calculated only considering throat pain or postoperative pain can interfere in the evaluation?

3. The kind of surgeries should be specified in the results.

4. The paper needs editing by a proficient English speaker for orthography, language and punctuation.

6. PLOS authors have the option to publish the peer review history of their article (what does this mean?). If published, this will include your full peer review and any attached files.

Reviewer #1: **Yes: **Mohammad Waheed El-Anwar

Reviewer #2: No

Reviewer #3: No

---

## [Author Response · Author response to Decision Letter 0]

15 Oct 2020

Dear, Editor，We have change the style of article and file name to meet PLOS ONE's style requirements. You can see the details in the revised article. We also change the file name about the article.

2) Thank you for submitting your clinical trial to PLOS ONE and for providing the name of the registry and the registration number. The information in the registry entry suggests that your trial was registered after patient recruitment began. PLOS ONE strongly encourages authors to register all trials before recruiting the first participant in a study.

(a) your reasons for your delay in registering this study (after enrolment of participants started);

In fact , we register on Feburary 28，2018,and recruited patients from June 12, 2018, to June 6, 2019. You can find the details on http://www.chictr.org.cn/searchproj.aspx by searching ChiCTR1800015007.1

In our protocol ，the time schedule is different because the preliminary experiment was made to make sure the technique is safe and useful, we apply to registered this study after that.

The authors confirm that all ongoing and related trials for this study are registered.

Please also ensure you report the date at which the ethics committee approved the study as well as the complete date range for patient recruitment and follow-up in the Methods section of your manuscript."

The details has been added in the article. We report the date at which the ethics committee approved the study as well as the complete date range for patient recruitment and follow-up in the Methods section. The ethics committee approved the study on Feburary 7，2018.

Because the syndrome is always self-healing， we follow-up the patients in 3days.

3) Please provide additional details regarding participant consent. In the ethics statement in the Methods and online submission information, please ensure that you have specified (1) whether consent was informed and (2) what type you obtained (for instance, written or verbal, and if verbal, how it was documented and witnessed). If your study included minors, state whether you obtained consent from parents or guardians.

The details has been added in the article. All patients gave written informed consent before participation in this study.

4) In your Methods section, please provide additional information about the participant recruitment method and the demographic details of your participants. Please ensure you have provided sufficient details to replicate the analyses such as: a) a description of how participants were recruited, and b) descriptions of where participants were recruited and where the research took place.

Participants were recruited for the study through advisement from Chancheng central hospital and the research was took place in PACU.

5) Please provide a sample size and power calculation in the Methods, or discuss the reasons for not performing one before study initiation.

We calculated that a sample size of 49 patients in each group would give the study 80% power to detect a 20-point greater improvement in VAS scores in the S group than in the I group at a two-sided significance level of 0.05.The calculation was operated on the website http://powerandsamplesize.com.The previous study has found that 90% of patients recovered after ultrasound guided injection of Isln,and 67.7% patients recovered after inhalation，so the proportion was used to calculated the sample size. http://powerandsamplesize.com/Calculators/Compare-2-Proportions/2-Sample-Equality?

6) To comply with PLOS ONE submission guidelines, in your Methods section, please provide additional information regarding your statistical analyses, including the name and version of the specific statistical software used for the analysis. For more information on PLOS ONE's expectations for statistical reporting, please see https://journals.plos.org/plosone/s/submission-guidelines.#loc-statistical-reporting.

The software named IBM SPSS statistics 24.0 was used for the analysis.

7) Please amend either the title on the online submission form (via Edit Submission) or the title in the manuscript so that they are identical.

We have changed the file names to make sure they are identical.

8) Please include captions for your Supporting Information files at the end of your manuscript, and update any in-text citations to match accordingly. Please see our Supporting Information guidelines for more information: http://journals.plos.org/plosone/s/supporting-information.

The supporting information files has been added to the bottom of the article, we have tried to revised the article follow the guideline.

9) We note that you have indicated that data from this study are available upon request. PLOS only allows data to be available upon request if there are legal or ethical restrictions on sharing data publicly. For information on unacceptable data access restrictions, please see http://journals.plos.org/plosone/s/data-availability#loc-unacceptable-data-access-restrictions.

The de-identified data sets has been uploaded this time following the guideline. Telephone number of ethics committee was 0757-827788715 

b) If there are no restrictions, please upload the minimal anonymized data set necessary to replicate your study findings as either Supporting Information files or to a stable, public repository 仓库知识库and provide us with the relevant URLs, DOIs, or accession numbers. Please see http://www.bmj.com/content/340/bmj.c181.long for guidelines on how to de-identify and prepare clinical data for publication. For a list of acceptable repositories, please see http://journals.plos.org/plosone/s/data-availability#loc-recommended-repositories.

We uploaded de-identified data sets as supplyment materials this time following the guideline.

---

## [Editor Report · Decision Letter 1]

22 Oct 2020

Ultrasound-guided Internal Branch of Superior Laryngeal Nerve Block on Postoperative Sore Throat: A Randomized Controlled Trial

PONE-D-20-15831R1

Dear Dr.Zhang Jinfang

We’re pleased to inform you that your manuscript has been judged scientifically suitable for publication and will be formally accepted for publication once it meets all outstanding technical requirements.

Kind regards,

Ehab Farag, MD FRCA FASA

Academic Editor

PLOS ONE
---

## [Editor Report · Acceptance letter]

27 Oct 2020

PONE-D-20-15831R1 

Ultrasound-guided Internal Branch of Superior Laryngeal Nerve Block on Postoperative Sore Throat: A Randomized Controlled Trial 

Dear Dr. Jinfang:

I'm pleased to inform you that your manuscript has been deemed suitable for publication in PLOS ONE. Congratulations! Your manuscript is now with our production department. 

Kind regards, 

on behalf of

Dr. Ehab Farag 

Academic Editor

PLOS ONE